# Shared Experience Actor-Critic for Multi-Agent Reinforcement Learning

**Filippos Christianos**
School of Informatics
University of Edinburgh
f.christianos@ed.ac.uk

**Lukas Schäfer**
School of Informatics
University of Edinburgh
l.schaefer@ed.ac.uk

**Stefano V. Albrecht**
School of Informatics
University of Edinburgh
s.albrecht@ed.ac.uk

## Abstract

Exploration in multi-agent reinforcement learning is a challenging problem, especially in environments with sparse rewards. We propose a general method for efficient exploration by sharing experience amongst agents. Our proposed algorithm, called *Shared Experience Actor-Critic* (SEAC), applies experience sharing in an actor-critic framework by combining the gradients of different agents. We evaluate SEAC in a collection of sparse-reward multi-agent environments and find that it consistently outperforms several baselines and state-of-the-art algorithms by learning in fewer steps and converging to higher returns. In some harder environments, experience sharing makes the difference between learning to solve the task and not learning at all.

## 1   Introduction

Multi-agent reinforcement learning (MARL) necessitates exploration of the environment dynamics and of the joint action space between agents. This is a difficult problem due to non-stationarity caused by concurrently learning agents and the fact that the joint action space grows exponentially in the number of agents [26]. The problem is exacerbated in environments with sparse rewards in which most transitions will not yield informative rewards.

We propose a general method for efficient MARL exploration by *sharing experience* amongst agents. Consider the simple multi-agent game shown in Figure 1 in which two agents must simultaneously arrive at a goal. This game presents a difficult exploration problem, requiring the agents to wander for a long period before stumbling upon a reward. When the agents finally succeed, the idea of sharing experience is appealing: both agents can learn how to approach the goal from two different directions after a successful episode by leveraging their collective experience. Such experience sharing facilitates a steady progression of all learning agents, meaning that agents improve at approximately equal rates as opposed to diverging in their learning progress. We show in our experiments that this approach of experience sharing can lead to significantly faster learning and higher final returns.

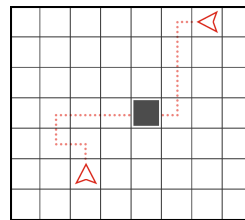

Figure 1: Two randomly-placed agents (triangles) must simultaneously arrive at the goal (square).

We demonstrate this idea in a novel actor-critic MARL algorithm, called *Shared Experience Actor-Critic* (SEAC).[1] SEAC operates similarly to independent learning [33] but updates the actor and critic parameters of an agent by combining gradients computed on the agent's experience with

weighted gradients computed on other agents' experiences. We evaluate SEAC in four sparse-reward multi-agent environments and find that it learns substantially faster (up to 70% fewer required training steps) and achieves higher final returns compared to several baselines, including: independent learning without experience sharing; using data from all agents to train a single shared policy; and MADDPG [20], QMIX [28], and ROMA [36]. Sharing experience with our implementation of SEAC increased running time by less than 3% across all environments compared to independent learning.

## 2 Related Work

**Centralised Training with Decentralised Execution:** The prevailing MARL paradigm of centralised training with decentralised execution (CTDE) [24, 28, 20] assumes a training stage during which the learning algorithm can access data from all agents to learn decentralised (locally-executable) agent policies. CTDE algorithms such as MADDPG [20] and COMA [12] learn powerful critic networks conditioned on joint observations and actions of all agents. A crucial difference to SEAC is that algorithms such as MADDPG only reinforce an agent's own tried actions, while SEAC uses shared experience to reinforce good actions tried by any agent, without learning the more complex joint-action critics. Our experiments show that MADDPG was unable to learn effective policies in our sparse-reward environments while SEAC learned successfully in most cases.

**Agents Teaching Agents:** There are approaches to leverage expertise of *teacher* agents to address the issue of sample complexity in training a *learner* agent [8]. Such teaching can be regarded as a form of transfer learning [25] among RL agents. The *teacher* would either implicitly or explicitly be asked to evaluate the behaviour of the *learner* and send instructions to the other agent. Contrary to our work, most such approaches do focus on single-agent RL [5, 11]. However, even in such teaching approaches for multi-agent systems [6, 7] experience is shared in the form of knowledge exchange following a teacher-learner protocol. Our approach shares agent trajectories for learning and therefore does not rely on the exchange of explicit queries or instructions, introducing minimal additional cost.

**Learning from Demonstrations:** Training agents from trajectories [30] of other agents [40] or humans [34] is a common case of teaching agents. Demonstration data can be used to derive a policy [15] which might be further refined using typical RL training [13] or to shape the rewards biasing towards previously seen expert demonstrations [4]. These approaches leverage expert trajectories to speed up or simplify learning for single-agent problems. In contrast, SEAC makes use of trajectories from other agents which are generated by concurrently learning agents in a multi-agent system. As such, we aim to speed up and synchronise training in MARL whereas learning from demonstrations focuses on using previously generated data for application in domains like robotics where generating experience samples is expensive.

**Distributed Reinforcement Learning:** Sharing experience among agents is related to recent work in distributed RL. These methods aim to effectively use large-scale computing resources for RL. Asynchronous methods such as A3C [22] execute multiple actors in parallel to generate trajectories more efficiently and break data correlations. Similarly, IMPALA [10] and SEED RL [9] are off-policy actor-critic algorithms to distribute data collection across many actors with optimisation being executed on a single learner. Network parameters, observations or actions are exchanged after each episode or timestep, respectively, and off-policy correction is applied. However, all these approaches only share experience of multiple actors to speed up learning of a single RL agent and by breaking correlations in the data rather than addressing synchronisation and sample efficiency in MARL.

**Population-play:** Population-based training is another line of research aiming to improve exploration and coordination in MARL by training a population of diverse sets of agents [35, 17, 16, 19]. Leibo et al. [17] note the overall benefits on exploration when sets of agents are dynamically evolved and mixed. In their work, some agents share policy networks and are trained alongside evolving sets of agents. Similarly to Leibo et al. [17], we observe benefits on exploration due to agents influencing each other's trajectories. However, SEAC is different from such population-play as it only trains a single set of distinct policies for all agents, thereby avoiding the significant computational cost involved in training multiple sets of agents.

# 3 Technical Preliminaries

**Markov Games:** We consider partially observable Markov games for $N$ agents [18]. A Markov game is defined by the tuple $(\mathcal{N}, \mathcal{S}, \{O^i\}_{i \in \mathcal{N}}, \{A^i\}_{i \in \mathcal{N}}, \mathcal{P}, \{R^i\}_{i \in \mathcal{N}})$, with agents $i \in \mathcal{N} = \{1, \dots, N\}$, state space $\mathcal{S}$, and joint action space $\mathcal{A} = A^1 \times \dots \times A^N$. Each agent $i$ only perceives local observations $o^i \in O^i$ which may depend deterministically or probabilistically on the current state. Function $\mathcal{P} : \mathcal{S} \times \mathcal{A} \mapsto \Delta(\mathcal{S})$ returns a distribution over successor states given a state and a joint action; $R^i : \mathcal{S} \times \mathcal{A} \times \mathcal{S} \mapsto \mathbb{R}$ is the reward function giving agent $i$'s individual reward $r^i$. Each agent $i$ seeks to maximise its discounted returns $G^i = \sum_{t=0}^{T} \gamma^t r_t^i$, with $\gamma$ and $T$ denoting the discount factor and total timesteps of an episode, respectively. $G_t^i$ denotes the returns for agent $i$ after timestep $t$.

In this work, we assume $O = O^1 = \dots = O^N$ and $A = A^1 = \dots = A^N$ in line with other recent works in MARL [31, 28, 12, 21]. (However, in contrast to these works we do not require that agents have identical reward functions, as will be discussed in Section 4.)

**Policy Gradient and Actor-Critic:** Policy Gradient (PG) algorithms are a class of model-free RL algorithms that aim to directly learn a policy $\pi_\phi$ parameterised by $\phi$, that maximises the expected returns. In REINFORCE [38], the simplest PG algorithm, this is accomplished by following the gradients of the objective $\nabla_\phi J(\phi) = \mathbb{E}_\pi [G_t \nabla_\phi \ln \pi_\phi(a_t|s_t)]$. Notably, the Markov property is not used, allowing the use of PG in partially observable settings. However, REINFORCE suffers from high variance of gradient estimation. To reduce variance of gradient estimates, actor-critic (AC) algorithms estimate Monte Carlo returns using a value function $V_\pi(s; \theta)$ with parameters $\theta$. In a multi-agent, partially observable setting, the simplest AC algorithm defines a policy loss for agent $i$

$$\mathcal{L}(\phi_i) = -\log \pi(a_t^i|o_t^i; \phi_i)(r_t^i + \gamma V(o_{t+1}^i; \theta_i) - V(o_t^i; \theta_i)) \tag{1}$$

with a value function minimising

$$\mathcal{L}(\theta_i) = ||V(o_t^i; \theta_i) - y_i||^2 \ \text{ with } \ y_i = r_t^i + \gamma V(o_{t+1}^i; \theta_i) \tag{2}$$

In practice, when $V$ and $\pi$ are parameterised by neural networks, sampling several trajectories in parallel, using $n$-step returns, regularisation, and other modifications can be beneficial [22]. To simplify our descriptions, our methods in Section 4 will be described only as extensions of Equations (1) and (2). In our experiments we use a modified AC algorithm as described in Section 5.3.

# 4 Shared Experience Actor-Critic

Our goal is to enable more efficient learning by sharing experience among agents. To facilitate experience sharing, we assume environments in which the local policy gradients of agents provide useful learning directions for all agents. Intuitively, this means that agents can learn from the experiences of other agents without necessarily having identical reward functions. Examples of such environments can be found in Section 5.

In each episode, each agent generates one on-policy trajectory. Usually, when on-policy training is used, RL algorithms only use the experience of each agent's own sampled trajectory to update the agent's networks with respect to Equation (1). Here, we propose to also use trajectories of other agents while considering that it is *off-policy* data, i.e. the trajectories are generated by agents executing different policies than the one optimised. Correcting for off-policy samples requires importance sampling. The loss for such off-policy policy gradient optimisation from a behavioural policy $\beta$ can be written as

$$\nabla_\phi \mathcal{L}(\phi) = -\frac{\pi(a_t|o_t; \phi)}{\beta(a_t|o_t)} \nabla_\phi \log \pi(a_t|o_t; \phi)(r_t + \gamma V(o_{t+1}; \theta) - V(o_t; \theta)) \tag{3}$$

In the AC framework of Section 3, we can extend the policy loss to use the agent's own trajectories (denoted with $i$) along with the experience of other agents (denoted with $k$), shown below:

$$\begin{aligned}
\mathcal{L}(\phi_i) = &- \log \pi(a_t^i|o_t^i; \phi_i)(r_t^i + \gamma V(o_{t+1}^i; \theta_i) - V(o_t^i; \theta_i)) \\
&- \lambda \sum_{k \neq i} \frac{\pi(a_t^k|o_t^k; \phi_i)}{\pi(a_t^k|o_t^k; \phi_k)} \log \pi(a_t^k|o_t^k; \phi_i)(r_t^k + \gamma V(o_{t+1}^k; \theta_i) - V(o_t^k; \theta_i))
\end{aligned} \tag{4}$$

---

**Algorithm 1** Shared Experience Actor-Critic Framework

---

**for** timestep $t = 1 \ldots$ **do**
    Observe $o_t^1 \ldots o_t^N$
    Sample actions $a_t^1, \ldots, a_t^N$ from $P(o_t^1; \phi_1), \ldots, P(o_t^N; \phi_N)$
    Execute actions and observe $r_t^1, \ldots, r_t^N$ and $o_{t+1}^1, \ldots, o_{t+1}^N$
    **for** agent $i = 1 \ldots N$ **do**
        Perform gradient step on $\phi_i$ by minimising Eq. (4)
        Perform gradient step on $\theta_i$ by minimising Eq. (5)
    **end for**
**end for**

---

Using this loss function, each agent is trained on both on-policy data while also using the off-policy data collected by all other agents at each training step. The value loss, in a similar fashion, becomes

$$\mathcal{L}(\theta_i) = ||V(o_t^i; \theta_i) - y_i^i||^2 + \lambda \sum_{k \neq i} \frac{\pi(a_t^k | o_t^k; \phi_i)}{\pi(a_t^k | o_t^k; \phi_k)} ||V(o_t^k; \theta_i) - y_k^i||^2$$

$$y_k^i = r_t^k + \gamma V(o_{t+1}^k; \theta_i)$$

(5)

We show how to derive the losses in Equations (4) and (5) for the case of two agents in Appendix C (generalisation to more agents is possible). The hyperparameter $\lambda$ weights the experience of other agents; we found SEAC to be largely insensitive to values of $\lambda$ and use $\lambda = 1$ in our experiments. A sensitivity analysis can be found in Appendix B. We refer to the resulting algorithm as *Shared Experience Actor-Critic* (SEAC) and provide pseudocode in Algorithm 1.

Due to the random weight initialisation of neural networks, each agent is trained from experience generated from different policies, leading to more diverse exploration. Similar techniques, such as annealing $\epsilon$-greedy policies to different values of $\epsilon$, have been observed [22] to improve the performance of algorithms.

It is possible to apply a similar concept of experience sharing to off-policy deep RL methods such as DQN [23]. We provide a description of experience sharing with DQN in Appendix D. Since DQN is an off-policy algorithm, experience generated by different policies can be used for optimisation without further considerations such as importance sampling. However, we find deep off-policy methods to exhibit rather unstable learning [14] compared to on-policy AC. We consider the generality of our method a strength, and believe it can improve other multi-agent algorithms (e.g. AC with centralised value function).

## 5 Experiments

We conduct experiments on four sparse-reward multi-agent environments and compare SEAC to two baselines as well as three state-of-the-art MARL algorithms: MADDPG [20], QMIX [28] and ROMA [36].

### 5.1 Environments

The following multi-agent environments were used in our evaluation. More detailed descriptions of these environments can be found in Appendix A.

**Predator Prey (PP), Fig. 2a:** First, we use the popular PP environment adapted from the Multi-agent Particle Environment framework [20]. In our sparse-reward variant, three predator agents must catch a prey by coordinating and approaching it simultaneously. The prey is a slowly moving agent that was pretrained with MADDPG and dense rewards to avoid predators. If at least two predators are adjacent to the prey, then they succeed and each receive a reward of one. Agents are penalised for leaving the bounds of the map, but otherwise receive zero reward.

**Starcraft Multi-Agent Challenge (SMAC), Fig. 2b:** The SMAC [29] environment was used in several recent MARL works [28, 12, 36]. SMAC originally uses dense reward signals and is primarily designed to test solutions to the multi-agent credit assignment problem. We present experiments on a

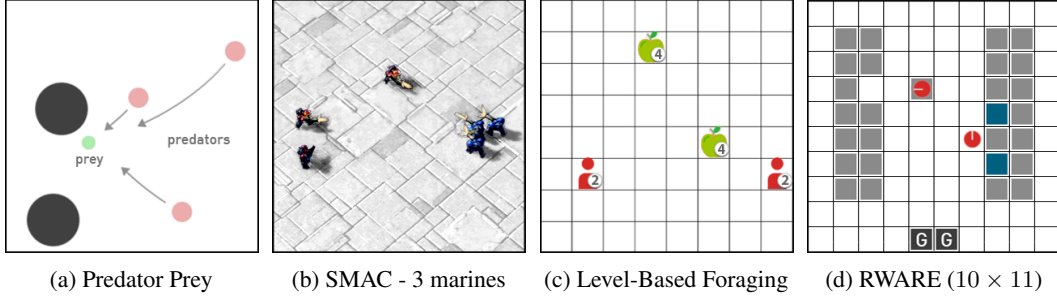

| (a) Predator Prey | (b) SMAC - 3 marines | (c) Level-Based Foraging | (d) RWARE ($10 \times 11$) |

Figure 2: Environments used in our evaluation. Controlled agents are coloured red.

simple variant that uses sparse rewards. In this environment, agents have to control a team of marines each represented by a single agent, to fight against an equivalent team of marines controlled by the game AI. With sparse rewards agents receive a single non-zero reward at the final timestep of each episode: a victory rewards 1, while a defeat $-1$.

**Level-Based Foraging (LBF), Fig. 2c:** LBF [1, 3] is a mixed cooperative-competitive game which focuses on the coordination of the agents involved. Agents of different skill levels navigate a grid world and collect foods by cooperating with other agents if required. Four tasks of this game will be tested, with a varied number of agents, foods, and grid size. Also, a cooperative variant will be tested. The reported returns are the fraction of items collected in every episode.

**Multi-Robot Warehouse (RWARE), Fig. 2d:** This multi-agent environment (similar to the one used in [2]) simulates robots that move goods around a warehouse, similarly to existing real-world applications [39]. The environment requires agents (circles) to move requested shelves (coloured squares) to the goal posts (letter 'G') and back to an empty location. It is a partially-observable collaborative environment with a very sparse reward signal, since agents have a limited view area and are rewarded only upon successful delivery. In the results, we report the total returns given by the number of deliveries over an episode of 500 timesteps on four different tasks in this environment.

## 5.2 Baselines

**Independent Actor-Critic (IAC):** We compare SEAC to independent learning [33], in which each agent has its own policy network and is trained separately only using its own experience. IAC uses an actor-critic algorithm for each agent, directly optimising Eqs. (1) and (2); and treating other agents as part of the environment. Arguably, independent learning is one of the most straightforward approaches to MARL and serves as reasonable baseline due to its simplicity.

**Shared Network Actor-Critic (SNAC):** We also compare SEAC to training a single shared policy among all agents. During execution of the environment, each agents gets a copy of the policy and individually follows it. During training, the sum of policy and value loss gradients are used to optimise the shared parameters. Importance sampling is not required since all trajectories are on-policy. Improved performance of our SEAC method would raise the question whether agents simply benefit from processing more data during training. Comparing against this baseline can also show that agents trained using experience sharing are not learning identical policies but instead learn different ones despite being trained on the same collective experience.

## 5.3 Algorithm Details

For all tested algorithms, we implement AC using n-step returns and synchronous environments [22]. Specifically, 5-step returns were used and four environments were sampled and passed in batches to the optimiser. An entropy regularisation term was added to the final policy loss [22], computing the entropy of the policy of each individual agent. Hence, the entropy term of agent $i$, given by $H(\pi(o_t^i; \phi_i))$, only considers its own policy. High computational requirements in terms of environment steps only allowed hyperparameter tuning for IAC on RWARE; all tested AC algorithms use the same hyperparameters (see Appendix B). All results presented are averaged across five seeds, with the standard deviation plotted as a shaded area.

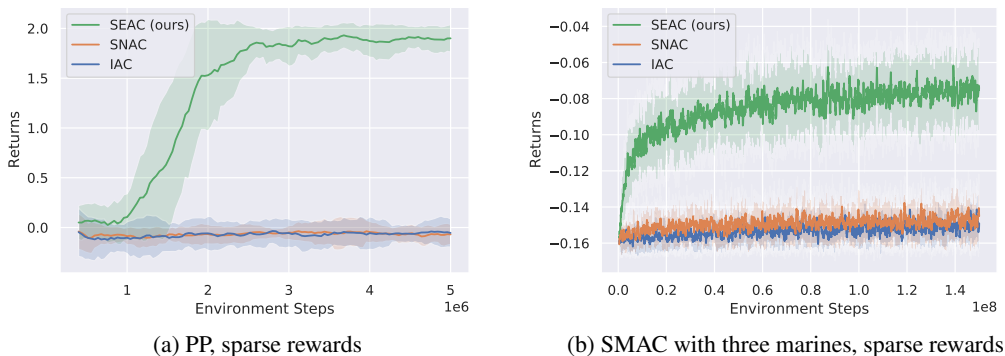

| (a) PP, sparse rewards | (b) SMAC with three marines, sparse rewards |

Figure 3: Mean training returns across seeds for sparse-reward variations of PP and SMAC-3m.

## 5.4 Results

Figures 3 to 5 show the training curves of SEAC, SNAC and IAC for all tested environments. For RWARE and LBF, tasks are sorted from easiest to hardest.

In the sparse PP task (Figure 3a) only SEAC learns successfully with consistent learning across seeds while IAC and SNAC are unable to learn to catch the prey.

In SMAC with sparse rewards (Figure 3b), SEAC outperforms both baselines. However, with mean returns close to zero, the agents have not learned to win the battles but rather to run away from the enemy. This is not surprising since our experiments (Table 1) show that even state-of-the-art methods designed for these environments (e.g. QMIX) do not successfully solve this sparsely rewarded task.

For LBF (Figure 4), no significant difference can be observed between SEAC and the two baseline methods IAC and SNAC for the easiest variant (Figure 4a) which does not emphasise exploration. However, as the rewards become sparser the improvement becomes apparent. For increased number of agents, foods and gridsize (Figures 4b to 4d), IAC and SNAC converge to significantly lower average returns than SEAC. In the largest grid (Figure 4c), IAC does not show any signs of learning due to the sparsity of the rewards whereas SEAC learns to collect some of the food. We also observe that SEAC tends to converge to its final policy in fewer timesteps than IAC.

In RWARE (Figure 5), results are similar to LBF. Again, the two baseline methods IAC and SNAC converge to lower average returns than SEAC for harder tasks. In the hardest task (Figure 5d), SEAC converges to final mean returns $\approx 70\%$ and $\approx 160\%$ higher than IAC and SNAC, respectively, and again converges in fewer steps than IAC.

In Table 1 we also present the final evaluation returns of three state-of-the-art MARL algorithms (QMIX [28], MADDPG [20], and ROMA [36]) on a selection of tasks. These algorithms show no signs of learning in most of these tasks. The only exceptions are MADDPG matching the returns of SEAC in the sparse PP task and QMIX performing comparably to SEAC in the cooperative LBF task. QMIX and ROMA assume tasks to be fully-cooperative, i.e. all agents receive the same reward signal. Hence, in order to apply the two algorithms, we modified non-cooperative environments to return the sum of all individual agent returns as the shared reward. While shared rewards could make learning harder, we also tested IAC in the easiest variant of RWARE and found that it learned successfully even with this reward setting.

We also evaluate *Shared Experience Q-Learning*, as described in Appendix D, and Independent Q-Learning [33] based on DQN [23]. In some sparse reward tasks, shared experience did reduce variance, and improved total returns. However, less impact has been observed through the addition of sharing experience to this off-policy algorithm compared to SEAC. Results can be found in Appendix D.

In terms of computational time, sharing experience with SEAC increased running time by less than $3\%$ across all environments compared to IAC. More details can be found in Appendix B.

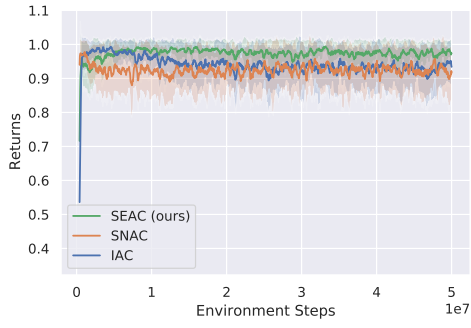

(a) LBF: $(12 \times 12)$, two agents, one food

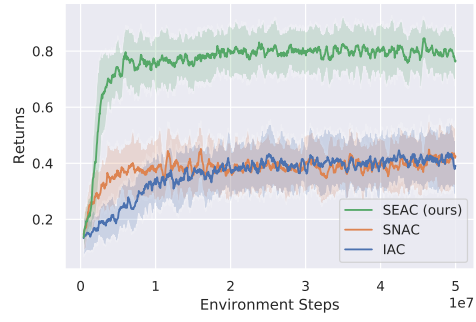

(b) LBF: $(10 \times 10)$, three agents, three foods

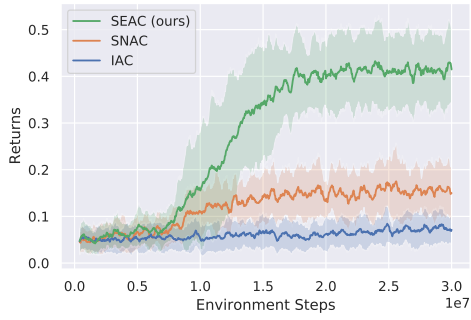

(c) LBF: $(15 \times 15)$, three agents, four food

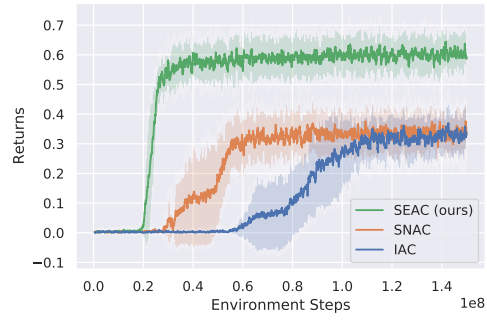

(d) LBF: $(8 \times 8)$, two agents, two foods, cooperative

Figure 4: Mean training returns across seeds on LBF. Tasks are sorted from easiest to hardest.

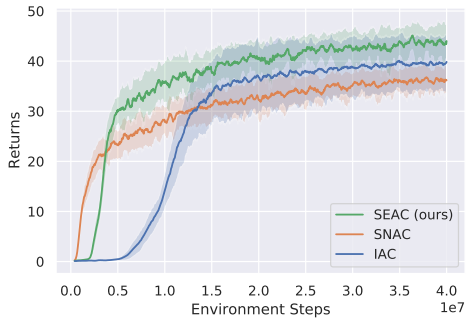

(a) RWARE: $(10 \times 11)$, four agents

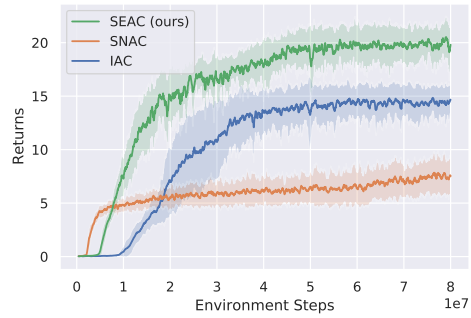

(b) RWARE: $(10 \times 11)$, two agents

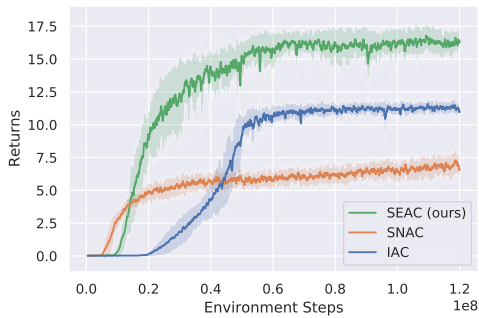

(c) RWARE: $(10 \times 11)$, two agents, hard

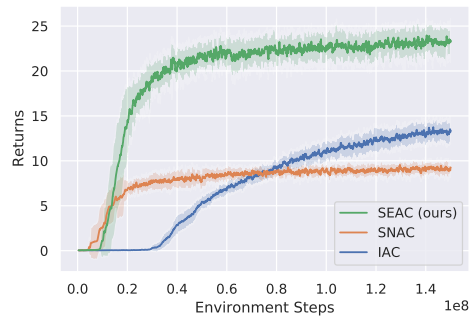

(d) RWARE: $(10 \times 20)$, four agents

Figure 5: Mean training returns across seeds on RWARE. Tasks are sorted from easiest to hardest.

Table 1: Final mean evaluation returns across five random seeds with standard deviation on a selection of tasks. Highest means per task (within one standard deviation) are shown in bold.

|  | IAC | SNAC | SEAC (ours) | QMIX | MADDPG | ROMA |
|---|---|---|---|---|---|---|
| PP (sparse) | -0.04 ±0.13 | -0.04 ±0.1 | **1.93 ±0.13** | 0.05 ±0.07 | **2.04 ±0.08** | 0.04 ±0.07 |
| SMAC-3m (sparse) | -0.13 ±0.01 | -0.14 ±0.02 | **-0.03 ±0.03** | **0.00 ±0.00** | -0.01 ±0.01 | **0.00 ±0.00** |
| LBF-(15x15)-3ag-4f | 0.13 ±0.04 | 0.18 ±0.08 | **0.43 ±0.09** | 0.03 ±0.01 | 0.01 ±0.02 | 0.03 ±0.02 |
| LBF-(8x8)-2ag-2f-coop | 0.37 ±0.10 | 0.38 ±0.10 | **0.64 ±0.08** | **0.79 ±0.31** | 0.01 ±0.02 | 0.01 ±0.02 |
| RWARE-(10x20)-4ag | 13.75 ±1.26 | 9.53 ±0.83 | **23.96 ±1.92** | 0.00 ±0.00 | 0.00 ±0.00 | 0.00 ±0.00 |
| RWARE-(10x11)-4ag | **40.10 ±5.60** | 36.79 ±2.36 | **45.11 ±2.90** | 0.00 ±0.00 | 0.00 ±0.00 | 0.01 ±0.01 |

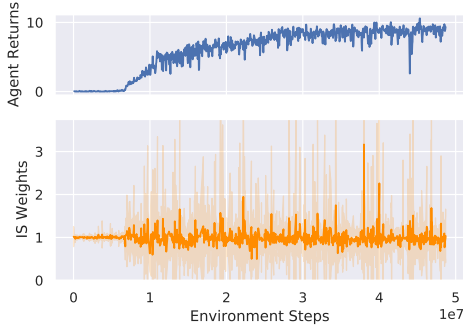

Figure 6: Importance weights of one SEAC agent in RWARE, $(10 \times 11)$, two agents, hard

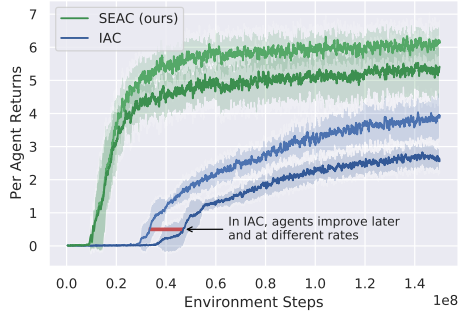

Figure 7: Best vs. Worst performing agents on RWARE, $(10 \times 20)$, four agents

## 5.5 Analysis

Similar patterns can be seen for the different algorithms across all tested environments. It is not surprising that IAC requires considerably more environment samples to converge, given that the algorithm is less efficient in using them; IAC agents only train on their own experience. This is further evident when noticing that in RWARE (Figs. 5a to 5d) the learning curve of SEAC starts moving upwards in roughly $1/N$ the timesteps compared to IAC, where $N$ refers to the number of agents. Also, it is not surprising that SNAC does not achieve as high returns after convergence: sharing a single policy across all agents impedes their ability to coordinate or develop distinct behaviours that lead to higher returns.

We conjecture that SEAC converges to higher final returns due to agents improving at similar rates when sharing experiences, combined with the flexibility to develop differences in policies to improve coordination. We observe that SEAC is able to learn similarly quickly to SNAC because the combined local gradients provide a very strong learning direction. However, while SNAC levels off at some point due to the use of identical policies, which limit the agents' ability to coordinate, SEAC can continue to explore and improve because agents are able to develop different policies to further improve coordination. Figure 6 shows that encountered importance weights during SEAC optimisation are centred around 1, with most weights staying in the range $[0.5; 1.5]$. This indicates that the agents indeed learn similar but not identical policies. The divergence of the policies is attributed to the random initialisation of networks, along with the agent-centred entropy factor (Section 5.3). The range of the importance weights also shows that, in our case, importance sampling does not introduce significant instability in the training. The latter is essential for learning since importance weighting for off-policy RL is known to suffer from significant instability and high variance through diverging policies [32, 27].

In contrast, we observe that IAC starts to improve at a much later stage than SEAC because agents need to explore for longer, and when they start improving it is often the case that one agent improves first while the other agents catch up later, which can severely impede learning. Figure 7 shows that agents using IAC end up learning at different rates, and the slowest one ends up with the lowest final returns. In learning tasks that require coordination, an agent being ahead of others in its training can impede overall training performance.

We find examples of agents learning at different rates being harmful to overall training in all our tested environments. In RWARE, an agent that learns to fulfil requests can make the learning more difficult for others by delivering all requests on its own. Agents with slightly less successful exploration have a harder time learning a rewarding policy when the task they need to perform is constantly done by others. In LBF, agents can choose to cooperate to gather highly rewarding food or focus on food that can be foraged independently. The latter is happening increasingly often when an agent is ahead in the learning curve as others are still aimlessly wandering in the environment. In the PP task, the predators must approach the prey simultaneously, but this cannot be the case when one predator does not know how to do so. In the SMAC-3m task, a single agent cannot be successful if its team members do not contribute to the fight. The agent would incorrectly learn that fighting is not viable and therefore prefer to run from the enemy, which however is not an optimal strategy.

## 6  Conclusion

This paper introduced SEAC, a novel multi-agent actor-critic algorithm in which agents learn from the experience of others. In our experiments, SEAC outperformed independent learning, shared policy training, and state-of-the-art MARL algorithms in ten sparse-reward learning tasks, across four environments, demonstrating improved sample efficiency and final returns. We discussed a theme commonly found in MARL environments: agents learning at different rates impedes exploration, leading to sub-optimal policies. SEAC overcomes this issue by combining the local gradients and concurrently learning similar policies for all agents, but it also benefits from not being restricted to identical policies, allowing for better coordination and exploration.

Sharing experience is appealing especially due to its simplicity. We showed that barely any additional computational power, nor any extra parameter tuning are required and no additional networks are introduced. Therefore, its use should be considered in all environments that fit the requirements.

Future work could aim to relax the assumptions made for tasks SEAC can be applied to and evaluate in further multi-agent environments. Also, our work focused on the application of experience sharing to independent actor-critic. Further analysis of sharing experience as a generally applicable concept for MARL and its impact on a variety of MARL algorithms is left for future work.

## Broader Impact

Multi-agent deep reinforcement learning has potential applications in areas such as autonomous vehicles, robotic warehouses, internet of things, smart grids, and more. Our research could be used to improve reinforcement learning models in such applications. However, it must be noted that real-world application of MARL algorithms is currently not viable due to open problems in AI explainability, robustness to failure cases, legal and ethical aspects, and other issues, which are outside the scope of our work.

That being said, improvements in MARL could lead to undue trust in RL models; having models that work well does not translate to models that are safe or which can be broadly used. Agents trained with these methods need to be thoroughly studied before being used in production. However, if these technologies are indeed used responsibly, they can improve several aspects of modern society such as making transportation safer, or performing jobs that might pose risks to humans.

## Funding Disclosure

This research was in part financially supported by the UK EPSRC Centre for Doctoral Training in Robotics and Autonomous Systems (F.C.), the Edinburgh University Principal's Career Development Scholarship (L.S.), and personal grants from the Royal Society and the Alan Turing Institute (S.A.).

## Footnotes

[1]We provide open-source implementations of SEAC in www.github.com/uoe-agents/seac and our two newly developed multi-agent environments: www.github.com/uoe-agents/lb-foraging (LBF) and www.github.com/uoe-agents/robotic-warehouse (RWARE).

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
