[Supplementary Material]



Figure 8: Three size variations of the multi-robot warehouse environment.

(a) Tiny size, two agents     (b) Small size, two agents     (c) Medium size, four agents

# A  Environments

## A.1  Multi-Robot Warehouse

The multi-robot warehouse environment (Figure 8) simulates a warehouse with robots moving and delivering requested goods. In real-world applications [38], robots pick-up shelves and deliver them to a workstation. Humans assess the content of a shelf, and then robots can return them to empty shelf locations. In this simulation of the environment, agents control robots and the action space for each agent is

$$A = \{\text{Turn Left, Turn Right, Forward, Load/Unload Shelf}\}$$

Agents can move beneath shelves when they do not carry anything, but when carrying a shelf, agents must use the corridors visible in Figure 8.

The observation of an agent consists of a $3 \times 3$ square centred on the agent. It contains information about the surrounding agents (location/rotation) and shelves.

At each time a fixed number of shelves $R$ is requested. When a requested shelf is brought to a goal location (dark squares in Fig. 8), another currently not requested shelf is uniformly sampled and added to the current requests. Agents are rewarded for successfully delivering a requested shelf to a goal location, with a reward of 1. A major challenge in this environments is for agents to deliver requested shelves but also afterwards finding an empty shelf location to return the previously delivered shelf. Agents need to put down their previously delivered shelf to be able to pick up a new shelf. This leads to a very sparse reward signal.

Since this is a collaborative task, as a performance metric we use the sum of the undiscounted returns of all the agents.

The multi-robot warehouse task is parameterised by:

- The size of the warehouse which is preset to either tiny ($10 \times 11$), small ($10 \times 20$), medium ($16 \times 20$), or large ($16 \times 29$).
- The number of agents $N$.
- The number of requested shelves $R$. By default $R = N$, but easy and hard variations of the environment use $R = 2N$ and $R = {}^{N}\!/_{2}$, respectively.

Note that $R$ directly affects the difficulty of the environment. A small $R$, especially on a larger grid, dramatically affects the sparsity of the reward and thus exploration: randomly bringing the correct shelf becomes increasingly improbable.

(a) Foraging-10x10-3p-3f   (b) Foraging-12x12-2p-1f   (c) Foraging-15x15-3p-4f   (d)   Foraging-8x8-2p-2f-coop

Figure 9: Four variations of level based foraging used in this work.

## A.2 Level-Based Foraging

The level-based foraging environment (Figure 9) represents a mixed cooperative-competitive game [1], which focuses on the coordination of the agents involved. Agents navigate a grid world and collect food by cooperating with other agents if needed.

More specifically, agents and food are randomly scattered in the grid world, and each is assigned a level. Agents can navigate in the environment and attempt to collect food placed next to them. The collection of food is successful only if the sum of the levels of all agents involved in collecting at the same time is equal to or higher than the level of the food. Agents are rewarded proportional to the level of food they took part in collecting. Episodes are terminated once all food has been collected or the maximum episode length of 25 timesteps is reached.

We are using full observability for this environment, meaning agents observe the locations and levels of all entities in the map. Each agent can attempt to move in all four directions and attempt to load adjacent food, for a total of five actions. After successfully loading a food, agents are rewarded:

$$r^i = \frac{FoodLevel * AgentLevel}{\sum FoodLevels \sum LoadingAgentsLevel}$$

This normalisation ensures that the sum of the agent returns on a solved episode equals to one.

Note that the final variant, Figure 9d, is a fully-cooperative environment. Food levels are always equal to the sum of all agents' levels, requiring all agents to load simultaneously, and thus sharing the reward.

## B Additional Experimental Details

Our implementations of IAC, SEAC, and SNAC closely follow A2C [21], using n-step returns and parallel sampled environments. Table 2 contains the hyperparameters used in the experiments. Hyperparameters for MADDPG, QMIX and ROMA were optimised using a grid search over learning rate, exploration rate and batch sizes with the grid centred on the hyperparameters used in the original papers and parameter performance tested in all used environments.

Table 3 contains process time required for running IAC and SEAC. Timings were measured on a $6^{th}$ Gen Intel i7 @ 4.6 Ghz running Python 3.7 and PyTorch 1.4. The average time for running and training on 100,000 environment iterations is displayed. Only process time (the time the program was active in the CPU) was measured, rounded to seconds. Often, the bottleneck is the environment and not the network update and as such, more complex and slower simulators, such as SMAC, show a lower percentage difference between algorithms.

Table 2: Hyperparameters used for implementation of SEAC, IAC and SNAC

| Hyperparameter | Value |
|---|---|
| learning rate | $3e^{-4}$ |
| network size | $64 \times 64$ |
| adam epsilon | 0.001 |
| gamma | 0.99 |
| entropy coef | 0.01 |
| value loss coef | 0.5 |
| GAE | False |
| grad clip | 0.5 |
| parallel processes | 4 |
| n-steps | 5 |
| $\lambda$ (Equations (4) and (5)) | 1.0 |

Table 3: Measured mean process time (mins:secs) required for 100,000 timesteps.

|  | IAC | SEAC | % increase |
|---|---|---|---|
| Foraging-10x10-3p-3f-v0 | 2:00 | 2:04 | 3.86% |
| Foraging-12x12-2p-1f-v0 | 1:22 | 1:24 | 2.94% |
| Foraging-15x15-3p-4f-v0 | 2:01 | 2:06 | 3.90% |
| Foraging-8x8-2p-2f-coop-v0 | 1:21 | 1:24 | 3.78% |
| rware-tiny-2ag-v1 | 1:41 | 1:43 | 1.65% |
| rware-tiny-2ag-hard-v1 | 2:05 | 2:09 | 2.97% |
| rware-tiny-4ag-v1 | 2:49 | 2:53 | 2.25% |
| rware-small-4ag-v1 | 2:50 | 2:55 | 2.44% |
| Predator Prey | 2:44 | 2:49 | 3.39% |
| SMAC (3m) | 6:23 | 6:25 | 0.38% |

(a) LBF ($15 \times 15$), 3 agents, 4 foods

(b) RWARE ($10 \times 20$), two agents

Figure 10: Training returns with different values of $\lambda$ in SEAC

Figure 10 shows the training returns with respect to different $\lambda$ values being applied in SEAC. We find that SEAC is not sensitive to tuning of the hyperparameter $\lambda$ with similar performance across a wide range of values. Much lower values for $\lambda$ closer to $0$ lead to decreased performance, eventually converging to IAC for $\lambda = 0$.

For calculation of evaluation returns (Table 1), the best saved models per seed were selected and evaluated for 100 episodes. During evaluation, QMIX and ROMA use $\epsilon = 0$, while MADDPG and AC algorithms apply stochastic policies.

## C  SEAC Loss Derivation

We provide the following derivation of SEAC policy loss, as shown in Equation (4), for a fully observable two-agent Markov game

$$\mathcal{M} = \left( \mathcal{N} = \{1, 2\}, \mathcal{S}, (A^1, A^2), \mathcal{P}, (R^1, R^2) \right)$$

As per Section 3, let $\mathcal{A} = A^1 \times A^2$ be the joint action space and $A = A^1 = A^2$.

In the following, we use $\pi_1$ and $\pi_2$ to denote the policy of agent 1 and agent 2 which are conditioned on parameters $\phi_1$ and $\phi_2$, respectively. We use $V^1$ and $V^2$ to denote the state value function of agents 1 and 2 which are conditioned on parameters $\theta_1$ and $\theta_2$.

In order to account for different action distributions under policies $\pi_1$ and $\pi_2$, we use importance sampling (IS) defined for any function $g$ over actions

$$\mathbb{E}_{a \sim \pi_1(a|s)} [g(a)] = \mathbb{E}_{a \sim \pi_2(a|s')} \left[ \frac{\pi_1(a|s)}{\pi_2(a|s')} g(a) \right]$$

which can be derived as follows

$$\mathbb{E}_{a \sim \pi_1(a|s)} [g(a)] = \int_a \pi_1(a|s) g(a) da = \int_a \frac{\pi_2(a|s')}{\pi_2(a|s')} \pi_1(a|s) g(a) da = \mathbb{E}_{a \sim \pi_2(a|s')} \left[ \frac{\pi_1(a|s)}{\pi_2(a|s')} g(a) \right]$$

**Assumption 1** (Reward Independence Assumption: A1). *We assume that an agent perceives the rewards as dependent only on its own action. Other agents are perceived as part of the environment.*

$$\forall s, s' \in \mathcal{S} : \forall a \in A : \hat{R}^1(s, a, s') = R^1(s, (a, \cdot), s')$$

$$\forall s, s' \in \mathcal{S} : \forall a \in A : \hat{R}^2(s, a, s') = R^2(s, (\cdot, a), s')$$

**Assumption 2** (Symmetry Assumption: A2). *We assume there exists a function $f : \mathcal{S} \mapsto \mathcal{S}$ such that*

$$\forall s, s' \in \mathcal{S} : \forall (a_1, a_2) \in \mathcal{A} : R^1(f(s), (a_2, a_1), f(s')) = R^2(s, (a_1, a_2), s')$$

$$and \ \forall s, s' \in \mathcal{S} : \forall (a_1, a_2) \in \mathcal{A} : \mathcal{P}(s, (a_1, a_2))(s') = \mathcal{P}(f(s), (a_2, a_1))(f(s'))$$

Intuitively, given a state $s$, $f(s)$ swaps the agents: agent 1 is in place of agent 2 and vice versa.

**Lemma 1** (Reward Symmetry: L1). *From these two assumptions, it follows that for any states $s, s' \in \mathcal{S}$, and any action $a \in A$ the following holds:*

$$\hat{R}^1(f(s), a, f(s')) = \hat{R}^2(s, a, s')$$

$$\hat{R}^2(f(s), a, f(s')) = \hat{R}^1(s, a, s')$$

*Proof.*

$$\hat{R}^1(f(s), a, f(s')) \overset{A1}{=} R^1(f(s), (a, \cdot), f(s')) \overset{A2}{=} R^2(s, (\cdot, a), s') \overset{A1}{=} \hat{R}^2(s, a, s')$$

$$\hat{R}^2(f(s), a, f(s')) \overset{A1}{=} R^2(f(s), (\cdot, a), f(s')) \overset{A2}{=} R^1(s, (a, \cdot), s') \overset{A1}{=} \hat{R}^1(s, a, s')$$

$\square$

During exploration, agent 1 and 2 follow policy $\pi_1$ and $\pi_2$, respectively. We will derive Equations (4) and (5) for training $\pi_1$ and $V^1$ using experience collected from agent 2. The derivation for optimisation of $\pi_2$ and $V^2$ using experience of agent 1 can be done analogously by substituting agent indices. Note that we only derive the off-policy terms of the SEAC policy and value loss. The on-policy terms of given losses are identical to A2C [21].

Agent 2 executes action $a_2$ in state $s$. Following Assumption 2, agent 1 needs to reinforce $\pi_1(a_2, f(s))$. Notably, in state $f(s)$, $a_1$ is sampled by $\pi_2$, so importance sampling is used to correct for this behavioural policy.

**Proposition 1** (Actor Loss Gradient).

$$\nabla_{\phi_1} \mathcal{L}(\phi_1) = \mathbb{E}_{a_2 \sim \pi_2} \left[ \frac{\pi_1(a_2|f(s))}{\pi_2(a_2|s)} \left( R^2(s, (\cdot, a_2), s') + \gamma V^1(f(s')) \right) \nabla_{\phi_1} \log \pi_1(a_2|f(s)) \right]$$

*Proof.*

$$\nabla_{\phi_1} \mathcal{L}(\phi_1) = \mathbb{E}_{\substack{a_1 \sim \pi_2 \\ a_2 \sim \pi_1}} \left[ Q^1(f(s), a_2) \nabla_{\phi_1} \log \pi_1(a_2|f(s)) \right]$$

$$\overset{IS}{=} \mathbb{E}_{a_1, a_2 \sim \pi_2} \left[ \frac{\pi_1(a_2|f(s))}{\pi_2(a_2|s))} Q^1(f(s), a_2) \nabla_{\phi_1} \log \pi_1(a_2|f(s)) \right]$$

$$= \mathbb{E}_{a_1, a_2 \sim \pi_2} \left[ \frac{\pi_1(a_2|f(s))}{\pi_2(a_2|s)} \left( R^1(f(s), (a_2, a_1), f(s')) + \gamma V^1(f(s')) \right) \nabla_{\phi_1} \log \pi_1(a_2|f(s)) \right]$$

$$\overset{A1}{=} \mathbb{E}_{a_2 \sim \pi_2} \left[ \frac{\pi_1(a_2|f(s))}{\pi_2(a_2|s)} \left( \hat{R}^1(f(s), a_2, f(s')) + \gamma V^1(f(s')) \right) \nabla_{\phi_1} \log \pi_1(a_2|f(s)) \right]$$

$$\overset{L1}{=} \mathbb{E}_{a_2 \sim \pi_2} \left[ \frac{\pi_1(a_2|f(s))}{\pi_2(a_2|s)} \left( \hat{R}^2(s, a_2, s') + \gamma V^1(f(s')) \right) \nabla_{\phi_1} \log \pi_1(a_2|f(s)) \right]$$

$$\overset{A1}{=} \mathbb{E}_{a_2 \sim \pi_2} \left[ \frac{\pi_1(a_2|f(s))}{\pi_2(a_2|s)} \left( R^2(s, (\cdot, a_2), s') + \gamma V^1(f(s')) \right) \nabla_{\phi_1} \log \pi_1(a_2|f(s)) \right]$$

$\square$

It should be noted that no gradient is back-propagated through the target $V^1(f(s'))$. In the same manner, the value loss, as shown in Equation (5), can be derived as follows.

**Proposition 2** (Value Loss)**.**

$$\mathcal{L}(\theta_1) = \mathop{\mathbb{E}}_{a_2 \sim \pi_2} \left[ \frac{\pi_1(a_2|f(s))}{\pi_2(a_2|s)} ||V^1(f(s)) - \left(R^2(s, (\cdot, a_2), s') + \gamma V^1(f(s'))\right)||^2 \right]$$

*Proof.*

$$\mathcal{L}(\theta_1) = \mathop{\mathbb{E}}_{\substack{a_1 \sim \pi_2 \\ a_2 \sim \pi_1}} \left[ ||V^1(f(s)) - \left(R^1(f(s), (a_2, a_1), f(s')) + \gamma V^1(f(s'))\right)||^2 \right]$$

$$\stackrel{IS}{=} \mathop{\mathbb{E}}_{a_1, a_2 \sim \pi_2} \left[ \frac{\pi_1(a_2|f(s))}{\pi_2(a_2|s)} ||V^1(f(s)) - \left(R^1(f(s), (a_2, a_1), f(s')) + \gamma V^1(f(s'))\right)||^2 \right]$$

$$\stackrel{A1}{=} \mathop{\mathbb{E}}_{a_2 \sim \pi_2} \left[ \frac{\pi_1(a_2|f(s))}{\pi_2(a_2|s)} ||V^1(f(s)) - \left(\hat{R}^1(f(s), a_2, f(s')) + \gamma V^1(f(s'))\right)||^2 \right]$$

$$\stackrel{L1}{=} \mathop{\mathbb{E}}_{a_2 \sim \pi_2} \left[ \frac{\pi_1(a_2|f(s))}{\pi_2(a_2|s)} ||V^1(f(s)) - \left(\hat{R}^2(s, a_2, s') + \gamma V^1(f(s'))\right)||^2 \right]$$

$$\stackrel{A1}{=} \mathop{\mathbb{E}}_{a_2 \sim \pi_2} \left[ \frac{\pi_1(a_2|f(s))}{\pi_2(a_2|s)} ||V^1(f(s)) - \left(R^2(s, (\cdot, a_2), s') + \gamma V^1(f(s'))\right)||^2 \right]$$

$\square$

## D  Shared Experience Q-Learning

### D.1  Preliminaries and Algorithm Details

**Deep Q-Networks:** Deep Q-Networks (DQNs) [22] are used to replace the traditional Q-tables [36] by learning to estimate Q-values with a neural network. The algorithm uses an experience (replay) buffer $D$, which stores all experience tuples collected, circumventing the issue of time-correlated samples. Also, due to the instability created by bootstrapping, a second network with parameters $\bar{\theta}$ is used and updated by slowly copying the parameters of the network, $\theta$, during training. The network is trained by minimising the loss

$$\mathcal{L}(\theta) = \frac{1}{M} \sum_{j=1}^{M} \left[ (Q(s_j, a_j; \theta) - y_j)^2 \right] \text{ with } y_j = r_j + \gamma \max_{a'} Q(s'_j, a'; \bar{\theta}) \tag{6}$$

computed over a batch of $M$ experience tuples $(s, a, r, s')$ sampled from $D$.

During each update of agent $i$, previously collected experiences are sampled from the experience replay buffer $D^i$ and used to compute and minimise the loss given in Equation (6). Independently

(a) RWARE ($10 \times 20$), two agents

(b) LBF: ($8 \times 8$), two agents, two fruits, cooperative

Figure 11: Average total returns of SEQL and IQL for RWARE and LBF tasks

applying DQN for each agent in a MARL environment is referred to as *Independent Q-Learning* (IQL) [32]. For such off-policy methods, sharing experience can naturally be done by sampling experience from either replay buffer $o, a, r, o' \sim D^1 \cup \ldots \cup D^N$ and using the same loss for optimisation. We refer to this variation of IQL as *Shared Experience Q-Learning* (SEQL). In our experiments, we sample the same number of experience tuples $\frac{M}{N}$ from each replay buffer and the same sampled experience samples are used to optimise each agent. Hence, SEQL and IQL are optimised using exactly the same number of samples, in contrast to SEAC and IAC.

## D.2 Results

Sharing experience in off-policy Q-Learning does improve performance, but does not show the same impact as for AC. We compare the performance of SEQL and IQL on one RWARE and LBF task to evaluate the impact of shared experience to off-policy MARL. Figure 11 shows the average total returns of SEQL and IQL on both tasks over five seeds. In the RWARE task, sharing experience appears to reduce variance considerably despite not impacting average returns significantly. On the other hand, on the LBF task average returns increased significantly by sharing experience and at its best evaluation even exceeded average returns achieved by SEAC. However, variance of off-policy SEQL and IQL is found to be significantly larger compared to on-policy SEAC, IAC and SNAC.