[Reviews · NeurIPS 2020]

Review 1

Summary and Contributions: Introduces a simple yet effective method which interpolates between self-play and independent MARL. This method outperforms baselines on challenging benchmark tasks with sparse reward.

Strengths: 1. General purpose method for fully cooperative MARL environments. 2. Well explained and parsimonious. 3. Convincing experiments on a range of environments from the literature.

Weaknesses: 1. Not clear how this method can be applied outside of fully cooperative settings, as the authors claim. The authors should justify this claim theoretically or empirically, or else remove it. 2. Missing some citations to set this in context of other MARL work e.g. recent papers on self-play and population-play with respect to exploration and coordination (such as https://arxiv.org/abs/1806.10071, https://arxiv.org/abs/1812.07019). 3. The analysis is somewhat "circumstantial", need more detailed experiments to be a convincing argument in this section. For example the claim in lines 235 - 236 seems to require further evidence to be completely convincing. 4. The link with self-play could be more clearly drawn out. As far as I can tell, the advantage of this over self-play is precisely the different initialization of the separate agents. It is surprising and important that this has such a significant effect, and could potentially spur a meta-learning investigation into optimal initialization for SEAC in future work.

Correctness: Yes, the methodology for the experiments and presentation of the results is rigorous.

Clarity: Yes, I was impressed by the writing quality of this work.

Relation to Prior Work: Yes, with a few caveats described in "weaknesses" above.

Reproducibility: Yes

Additional Feedback: See "weaknesses" for improvement suggestions. I found this paper to be an enjoyable read. Response to authors: many thanks for your replies to my comments. It would be useful for you to include a sentence indicating the situations in which your algorithm may and may not be applied in your introduction, so that practitioners may easily see whether it is relevant to their setting. My view remains that this is a good paper.


Review 2

Summary and Contributions: Paper presents a novel method for multi-agent RL called Shared Experience Actor-Critic (SEAC). The authors demonstrate that this method outperforms state-of-the-art multi-agent RL algorithms as well as baselines. This is achieved by sharing experiences between agents while still having the polices to converge to different parameters sets (weights). The another strength is the simplicity of the algorithm making it more likely to be widely adopted by the community. Results are compared with baselines and state-of-the-art multi-agent RL algorithm and presented with standard deviation.

Strengths: The main results are impressive where the SEAC algorithm achieves higher score than two state-of-the-art multi-agent algorithms like MADDPG. Baselines are well designed and adds to highlight the strength of MARL over simpler solutions.

Weaknesses: Testing RL algorithms with limited resources for testing different hyperparameters always causes some issues since RL has been shown to very sensitive to hyperparameters, see for example Deep Reinforcement Learning that Matters (Peter Henderson et. al. 2017). This means that when comparing to algorithm there is no guarantee that the same architecture and hyperparameters will be optimal for both SEAC and MADDPG. Comparison to later MARL algorithms such as "Actor-Attention-Critic for Multi-Agent Reinforcement Learning" from ICML 2019 is missing and not cited. In their paper they show that they out-perfrom MADDPG and that their algorithm is able to handle non-cooperative settings.

Correctness: Yes, claims are correct as far as I can tell. I had a look at the derivations in the supplementary material and could not find any issues.

Clarity: Paper is very well-written and easy to follow. There are very few typos. Plots are also easy to read and show standard deviation as well as mean for the resuls.

Relation to Prior Work: Some recent prior work is missing, as noted in the weakness section, otherwise the related work section is well-written and gives a clear picture.

Reproducibility: Yes

Additional Feedback:


Review 3

Summary and Contributions: In this work, the multi-agent reinforcement learning algorithm with sharing experiences among agents is proposed, where homogeneous agents are assumed. We may put this method as Centralized Training with Decentralized Execution (CTDE) due to the experience sharing. Based on the simple off-policy -- which comes from using other agents’ policy -- actor-critic updates (eq (4) and (5)), each agent updates its own networks. Their contribution is on the empirical results which show SEAC outperforms their baselines including Independent Actor-Critic (IAC), Shared Network Actor-Critic (SNAC), QMIX and MADDPG. Cooperative tasks were basically considered.

Strengths: Empirical tests were done over multiple environments and evaluations were done in various perspectives.

Weaknesses: Weaker baselines seem to be used.

Correctness: The empirical methodology seems to be improved.

Clarity: The submission is clearly written.

Relation to Prior Work: The relations are clearly discussed.

Reproducibility: Yes

Additional Feedback: I like authors tried their experiments in various perspectives, but experience sharing is occasionally seen from the existing literature. For example, although it wasn’t mentioned in the paper, [1] used experience sharing among agents for their implementation, and I believe there may be other works with the topic of “MARL for homogeneous agents”. The main reason I score “below acceptance” is that quite weak baselines seem to be used: - In Table 1, QMIX and MADDPG highly underperforms SEAC and other baselines (IAC, SNAC). However, since methods with CTDE are mostly more stable than independent learning methods, I think this part should be explained in more detail. - In addition, both QMIX and MADDPG can use experience sharing by exploiting the homogeneity of agents. (For 3 agents, e.g., a set observation (O1, O2, O3) can be reused with permutation (O1, O3, O2), (O2, O1, O3), (O2, O3, O1).) But, it seems like QMIX and MADDPG don’t exploit such properties. [References] [1] Song, Jiaming, et al. "Multi-agent generative adversarial imitation learning." Advances in neural information processing systems. 2018. ---------- After I read through other reviewers' comments and authors' responses, but I decided to keep my score (below the acceptance threshold) for the following reasons. Although other reviewers have argued the strength of this work from the importance weighting and simplicity of methods, I still think there should have been stronger baselines. Since using shared experience among agents implies there exists a sort of centralized coordination, I think CTDE for homogeneous agents with shared experience should have been considered as their baselines, which was already considered in MACK (from MA-GAIL/MA-AIRL papers [1, 2]). I also think solving sparse environments with the multi-step return is not a novel idea since MACK [1, 2] used the Generalized Advantage Estimation (GAE) and can definitely solve such an issue. Regarding the same issue, there’s also a previous work IGASIL [4] *without* using a centralized critic, which hasn’t been discussed from the submission as well. In addition, the authors’ feedback for my review was more on the scalability issue of CTDE, but I think this is also quite well-solved by [5 for homogeneous agents, 6 for general cases by using a critic network with shared weights]. [1] Song, Jiaming, et al. "Multi-agent generative adversarial imitation learning." Advances in neural information processing systems. 2018. [2] Yu, Lantao, Jiaming Song, and Stefano Ermon. "Multi-agent adversarial inverse reinforcement learning." arXiv preprint arXiv:1907.13220 (2019). [3] Yang, Yaodong, et al. "Mean field multi-agent reinforcement learning." arXiv preprint arXiv:1802.05438 (2018). [4] Hao, Xiaotian, et al. "Independent generative adversarial self-imitation learning in cooperative multiagent systems." arXiv preprint arXiv:1909.11468 (2019). [5] Yang, Y., Luo, R., Li, M., Zhou, M., Zhang, W., & Wang, J. (2018). Mean field multi-agent reinforcement learning. arXiv preprint arXiv:1802.05438. [6] Iqbal, Shariq, and Fei Sha. "Actor-attention-critic for multi-agent reinforcement learning." International Conference on Machine Learning. 2019.


Review 4

Summary and Contributions: The authors describe an actor-critic algorithm for multi-agent RL that leverages experience sharing across homogeneous agents for improved sample efficiency. This Shared Experience Actor-Critic (SEAC) method is theoretically well-grounded and evaluated across a broad set of environments, including two new environments that the authors intend to publicly release.

Strengths: Overall this paper is of high quality. Exploration in RL is a core problem of relevance to the NeurIPS community, and leveraging a specific property of the multi-agent setting (global access to other agents' experiences) to improve sample efficiency is a meaningful contribution. The authors provide a good description and derivation of the method, and go to considerable effort to position it in the context of related work. Their proposed SEAC algorithm is refreshingly simple and trivial to implement, and yet achieves convincing performance across a broad set of MARL tasks and compared to a representative set of prior algorithms. Their experimental analysis is sound and well-described, and I have confidence in the accuracy of their findings.

Weaknesses: Two relatively minor concerns that I'd like to see discussed by the authors. I'm inclined to increase my score if there are good explanations for both points. My understanding of this paper is that the core innovation is to repurpose importance weighting as a method of off-policy correction, typically used to compensate for gradient staleness in distributed on-policy agents, to instead compensate for the different likelihood of actions under different agents' policies. This is an elegant idea and leads to an algorithm that is both simple and yields convincing results. To be honest, I'm surprised that this works. The likelihood ratios are going to tend to zero very quickly unless the agent policies are very similar (suboptimal behavior for most MA tasks by design), and in that scenario other agents' experiences are going to have negligible effect on the gradient update. But my surprise is not a good measure of research quality, and the results appear to disagree with me. It might be worth discussing why my intuition is wrong. I'd also like to see a discussion on what would be needed to extend SEAC to a regime of non-homogeneous agents (i.e. different objectives). I got excited when I saw the PP results, but the prey was using a pre-trained network so this is cheating in some respect. For context, it seems to me that SEAC is most closely related to MADDPG, where agent experiences are pooled in replay memory and sampled off-policy with agent-specific policies and critics. The authors are correct in identifying that MADDPG agents do not explicitly learn from the (S,A) -> R experience of other agents. However, I have two observations on this topic. First, this type of experience sharing only works in the regime where agents are homogeneous (same capabilities and objective), whereas MADDPG allows for pooling of experiences under heterogeneous policies. Unless I'm mistaken, SEAC does not work in these regimes, so in this particular sense MADDPG is more general. Second, this "limitation" of MADDPG is actually a design decision. One could trivially modify MADDPG to symmetrically update every agent based on every other agent's experience (again assuming homogeneity), but this essentially collapses to maintaining a single global critic, which IIUC the MADDPG were explicitly trying to avoid based on the observation that separate critics lead to better performance.

Correctness: Yes

Clarity: Yes

Relation to Prior Work: Yes

Reproducibility: Yes

Additional Feedback: UPDATE I thank the authors for their effort in clarifying my concerns. Overall, I think this is a good submission. The explanation re: importance weights is satisfying, and the idea that this method of experience sharing will lead to similar but different-enough policies to be useful is an interesting one. The nature of the proposed algorithm is that it will *only* lead to similar policies, and I think this limitation is worth stating more explicitly (along with citations supporting the related claim that this is okay). I agree with the other reviewers that direct evaluation to more recent methods would be ideal, but perhaps not strictly necessary for acceptance. For this reason I am leaving my initial overall score (7) unchanged, but have increased my confidence score as a result of thoughtful response.

[Author Response · NeurIPS 2020]

We thank the reviewers for their thoughtful feedback and suggestions. We are encouraged to read that they found our work clearly written (R1, R2, R3, R4) and they appreciated the simplicity (R2, R4), generality (R1) as well as convincing performance (R1, R2, R4) of our method.

**Reviewer 1**

**3.1** The fully-cooperative setting assumes that agents share the same rewards in each time step. In contrast, we allow agents to use individual reward functions (line 79), but assume that the environment is such that the local policy gradients of agents provide useful learning directions for all agents (lines 99-103). Intuitively, this means that rewards are in a sense symmetrical between agents, in that actions that work for one agent also work for another agent if they swapped positions. Hence, SEAC could also be applied in some competitive tasks. We use examples of such environments in our evaluation. For example, most LBF tasks are not fully cooperative (Fig. 3efg) in the sense that agents do not share rewards. A food captured by a single agent can deprive other agents of rewards (competition). We provide a theoretical grounding of our symmetry assumption in Section C of the supplementary material.

**3.2 & 3.4** We agree that the citations on self- and population-play, especially with respect to exploration, are relevant. We will investigate this relation further and include respective related work in Section 2.

**Reviewer 2**

**3.1** Hyperparameters for MADDPG and QMIX were optimised using a grid search over learning rate, exploration rate, batch sizes (and more) with the grid centred on the hyperparameters used in the original papers and parameter performance tested in all used environments. Thus, MADDPG/QMIX used individually-tuned hyperparameters and were not limited to the hyperparameters and network architectures tuned on the SEAC/SNAC/IAC algorithms. This clarification was missing from the submitted version, and we now include it in Section 5.3.

**3.2** We compared against MADDPG and QMIX (as well as IAC/SNAC baselines) which, in our experience, show good performance across different tasks. We will look into the cited work from ICML 2019.

**Reviewer 3**

**8.** *[Why CDTE algorithms underperform SEAC and baselines]* MADDPG and QMIX rely on the global state during training, which is not always be desirable: This global state (often approximated through a concatenation of agents' observations) grows with respect to the number of agents, which leads to large networks that are harder to train, especially in the absence of a dense reward signal. This is apparent in RWARE, where observations are high-dimensional and rewards are very sparse, and both QMIX and MADDPG do not learn efficiently (Tab. 1). We offer an alternative CTDE approach, SEAC, in which the centralised training is exploited without growing network sizes.

**8.** For a fair comparison we remain consistent with the original implementations of QMIX and MADDPG. Indeed, MADDPG/QMIX and other methods could also use experience sharing. We consider this generality a strength of our approach (e.g. see discussion in lines 124-130). Our experiments on methods using experience replay (Section D of supplementary material) also indicated a meaningful improvement, although not as significant as for SEAC.

**8.** Thank you for pointing us to the cited paper which we now include in the related work section.

**Reviewer 4**

**3.Q1** Indeed, the likelihood ratios could tend to zero which can zero-out the respective gradients. However, our experiments indicate that the IS weights stay in a desirable range (Fig 5: [IS weights centered around 1.0+-0.5]) indicating that agents learn similar but not identical policies (lines 226-227). We observe that *small* differences in policies can have a significant impact in the overall coordination of agents in the tested environments. In RWARE (Fig. 2a), optimal behaviours are very similar (when seeing a requested shelf, pick it up and deliver it) but small differences, e.g. some agents giving way to others in corridors, allow for a much better overall efficiency. Using identical policies (SNAC) leads to agents colliding and clustering in narrow places, disrupting each other. Similarly, in LBF (Fig. 2c) agents must learn similar policies (go towards the same food locations) but their policies must collect the food from different neighbouring cells to avoid collision with other agents. In that sense, the reviewer's intuition is not wrong, but we consider environments in which agents can coordinate effectively without requiring greatly dissimilar policies.

**3.Q2** We agree that MADDPG is more general in this regard, however we would like to point out that SEAC allows for some flexibility with experience sharing beyond fully-homogeneous tasks as long as our symmetry assumption holds between some agents (see answer to Reviewer 1, 3.1). For example, learning in a predator-prey environment with multiple preys and multiple predators could make use of SEAC by sharing experience only within preys and predators, respectively. This could be trivially achieved by setting the lambdas to 1 for agents in the same role, and 0 for other agents. Learning lambda values as part of the RL process could be the subject of future work, and might strongly relate to prior work in MARL with role assignment and similar ideas.

[Meta-Review · NeurIPS 2020]

This paper introduces a simple idea for MARL, using importance weights to correct for off-policy. Generally, the reviewers agree that the paper is clear and well written. Although the main idea is very natural and intuitive, as pointed out by reviewer 4, it is not intuitive that is would actually work. Therefore, one of the strengths of this paper is to show that intuition fails us in this case. The reviewers point out some weaknesses in the empirical sections, in particular comparisons with other methods, and we hope that the authors will be able to address some of these in the final version of the paper.